# Electrochemical C–H phosphorylation of arenes in continuous flow suitable for late-stage functionalization

Hao Long[1], Chong Huang[1], Yun-Tao Zheng[1], Zhao-Yu Li[1], Liang-Hua Jie[1], Jinshuai Song [2], Shaobin Zhu [3] & Hai-Chao Xu [1✉]

The development of efficient and sustainable methods for carbon-phosphorus bond formation is of great importance due to the wide application of organophosphorus compounds in chemistry, material sciences and biology. Previous C–H phosphorylation reactions under nonelectrochemical or electrochemical conditions require directing groups, transition metal catalysts, or chemical oxidants and suffer from limited scope. Herein we disclose a catalyst- and external oxidant-free, electrochemical C–H phosphorylation reaction of arenes in continuous flow for the synthesis of aryl phosphorus compounds. The C–P bond is formed through the reaction of arenes with anodically generated P-radical cations, a class of reactive intermediates remained unexplored for synthesis despite intensive studies of P-radicals. The high reactivity of the P-radical cations coupled with the mild conditions of the electrosynthesis ensures not only efficient reactions of arenes of diverse electronic properties but also selective late-stage functionalization of complex natural products and bioactive compounds. The synthetic utility of the electrochemical method is further demonstrated by the continuous production of 55.0 grams of one of the phosphonate products.

[1] State Key Laboratory of Physical Chemistry of Solid Surfaces, Key Laboratory of Chemical Biology of Fujian Province, and College of Chemistry and Chemical Engineering, Xiamen University, 361005 Xiamen, China. [2] Green Catalysis Center, College of Chemistry, Zhengzhou University, 450001 Zhengzhou, China. [3] NanoFCM INC., Xiamen Pioneering Park for Overseas Chinese Scholars, 361006 Xiamen, China. ✉email: haichao.xu@xmu.edu.cn

Aryl phosphorus compounds have wide applications in medicinal chemistry[1], material science[2], and catalysis as ligands and Lewis acid catalysts[3,4]. In addition, brigatinib, which contains a phenylphosphine oxide motif and serves as an anaplastic lymphoma kinase (ALK) inhibitor, has been achieved commercial success for treating metastatic non-small-cell lung cancer (NSCLC)[5]. Compared with the well-established Hirao reaction, which involves transition metal-catalyzed cross coupling of arylhalides with a P-donor[6–10], C–H phosphorylation is more attractive and minimizes substrate prefunctionalization. In this context, several transition-metal catalyzed, directing group-assisted arene C–H phosphorylation reactions have been reported (Fig. 1a)[11–13]. The use of directing groups limits the scope of the arenes. Non-directed C(aryl)–H phosphorylation has also been reported and usually proceeds through a radical mechanism (Fig. 1b). In these reactions, either the arene or the phosphorus coupling partner is oxidized under photochemical[14,15] or transition-metal catalyzed[16,17] conditions to a radical intermediate, which then reacts with the other coupling partner to achieve C–P bond formation. These radical-based reactions are generally limited to electron-rich arenes or azoles[18–21] due to the need to oxidize the arene to a radical cation and the moderate reactivity of the neutral P-radicals. In addition, excess arenes are frequently employed for the radical C–H phosphorylation reactions.

Organic electrochemistry has been shown to be an enabling technology in addressing the challenges in sustainable synthesis[22–37]. In this context, electrochemical C–H phosphorylation of electron-rich heteroaromatics[38,39] and arenes bearing directing groups[40,41] have been reported. Reactions of electron-deficient arenes require metal catalysts and impractical three-electrode configuration in divided cells[42–47].

Inspired by the observation that protonation of aminyl radicals to aminium radicals dramatically increases their reactivity toward π-systems[48,49], we envision that the P-radical cation should be more reactive than the P-radicals and may react with electron-deficient arenes (Fig. 1c). The challenge is that the reactivity of P-radical cations toward electron-deficient arenes remains unknown and the trialkyl phosphite radical cations are known to react facilely with trialkyl phosphite precursor to form dimer

radical cations[50,51]. Our interests in electrochemically driven radical reactions prompted us to investigate the reactivity of P-radical cations[52]. We surmise that the use of a continuous flow electrochemical reactor[53–68] should be helpful in facilitating P-radical cation formation and its subsequent trapping reaction by the arenes because of the large ratio of electrode surface area to volume and efficient mass transfer in the flow reactor.

Herein we disclose a catalyst- and external oxidant-free, electrochemical C–H phosphorylation reaction of arenes in continuous flow for the synthesis of aryl phosphorus compounds (Fig. 1d). The electrochemical method is characterized by a broad scope, good functional group tolerance, easy scale-up, and compatibility with late-stage C–H functionalization of derivatives of complex natural products and drug molecules.

## Results

**Reaction development.** The C–H phosphorylation of benzoate 1 with triethyl phosphite [P(OEt)$_3$] was chosen as a model reaction for the optimization of electrolysis conditions (Table 1 and Supplementary Table 1). The electrolysis was conducted in an undivided continuous flow electrochemical cell equipped with a graphite anode, a Pt cathode, and a flow channel cut in a fluorinated ethylene propylene spacer with 0.25 mm thickness (see the Supplementary Methods for details)[62]. An optimal 70% yield of phosphonate 2 was obtained when a solution of 1 (1 equiv), P(OEt)$_3$ (5 equiv), HBF$_4$ (2 equiv) and H$_2$O (2 equiv) in MeCN was passed through the flow cell at 0.2 mL min$^{-1}$. The C–H functionalization occurred exclusively at the position ortho to the ester group because the steric hindrance of tBu group prevented reaction at its ortho position. HBF$_4$ (entry 2) and H$_2$O (entry 3) were critical for success as 2 was not formed without either one. Increasing the amount of P(OEt)$_3$ to 7 equiv (entry 4) or decreasing to 3 equiv (entry 5) or replacing P(OEt)$_3$ with diethyl phosphite [HPO(OEt)$_2$] (entry 6) were all detrimental to the formation of 2. A high concentration of P(OEt)$_3$ probably led to competitive trapping of the P-radical cation with P(OEt)$_3$. The use of other acids such as TFA (entry 7), AcOH (entry 8), or TfOH (entry 9), or Lewis acid Sc(OTf)$_3$ (entry 10) instead of HBF$_4$ resulted in either no product formation or low yield and

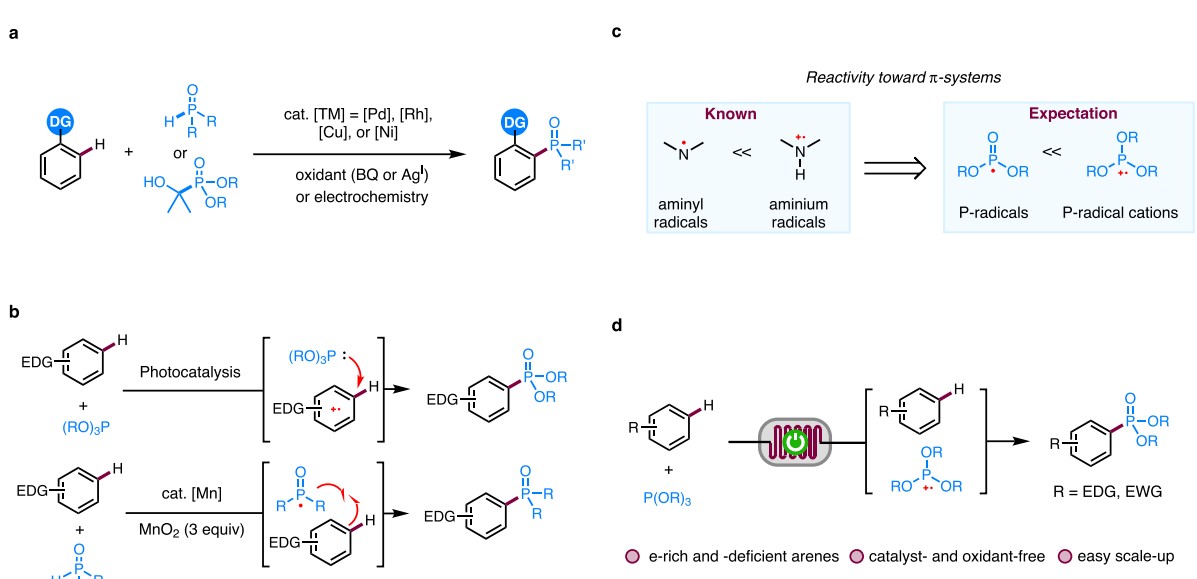

**Fig. 1 C–H phosphorylation of arenes. a** Transition metal-catalyzed directed arene C–H phosphorylation. **b** Radical-mediated C–H phosphorylation of arenes. **c** Envisioned strategy for increasing the reactivity of P-radicals. **d** Electrochemical arene C–H phosphorylation in continuous flow (this work). DG directing group. BQ *p*-benzoquinone. EDG electron-donating group. EWG electron-withdrawing group.

**Table 1 Optimization of reaction conditions.**

| Entry | Deviation from standard conditions | Yield of 2 (%)[a] |
|---|---|---|
| 1 | None | 78, 70[b] |
| 2 | No HBF$_4$•Et$_2$O | 0 (85) |
| 3 | No H$_2$O | 0 (72) |
| 4 | P(OEt)$_3$ (7 equiv) | 18 (70) |
| 5 | P(OEt)$_3$ (3 equiv) | 0 (83) |
| 6 | HPO(OEt)$_2$ instead of P(OEt)$_3$ | 0 (90) |
| 7 | TFA (2 equiv) instead of HBF$_4$•Et$_2$O | 0 (90) |
| 8 | AcOH (2 equiv) instead of HBF$_4$•Et$_2$O | 0 (90) |
| 9 | TfOH (2 equiv) instead of HBF$_4$•Et$_2$O | 30 (50) |
| 10 | Sc(OTf)$_3$ (0.3 equiv) instead of HBF$_4$•Et$_2$O | 0 (80) |
| 11[c] | Reaction in batch | 36 (52) |

*TFA* trifluoroacetic acid, *AcOH* acetic acid, *TfOH* triflic acid, *Sc(OTf)$_3$* scandium(III) triflate.
Standard conditions: graphite anode (10 cm$^2$), Pt cathode, interelectrode distance (0.25 mm), **1** (0.2 mmol), P(OEt)$_3$ (1.0 mmol), HBF$_4$•Et$_2$O (0.4 mmol), H$_2$O (0.4 mmol), MeCN (4 mL), flow rate = 0.2 mL min$^{-1}$, $t_r$ (calculated residence time) = 75 s, constant current (55 mA), 3.4 F mol$^{-1}$.
[a]Yield determined by $^1$H-NMR analysis using 1,3,5-trimethoxybenzene as the internal standard. Unreacted **1** in parenthesis.
[b]Isolated yield.
[c]Graphite anode, Pt cathode, 5.5 mA, 3.4 F mol$^{-1}$.

recovery of most **1**. Reaction in a batch reactor with the same amount of charge afforded the desired **2** in 36% yield (entry 11).

**Evaluation of substrate scope**. Under the optimal reaction conditions, the electrochemical C–H phosphorylation with P(OEt)$_3$ tolerated arenes of various electronic properties (Fig. 2), including the highly electron-deficient dimethyl terephthalate (**4**) and dimethyl phthalate (**5**). The site selectivity of the C–P bond formation was like Friedel-Crafts reactions and favored reaction at ortho- or para- to the electron-releasing groups if there were no severe steric hindrance. Besides benzene and its derivatives, 2-substituted thiophenes reacted successfully at 4-position to give the corresponding products (**29** and **30**). Trialkyl phosphites bearing primary alkyl chains of different lengths reacted smoothly to give the corresponding phosphonate products (**31–36**). The electrochemical method was also suitable for late-stage functionalization of relatively complex natural products and bioactive compounds (**37–46**). Compounds that contained multiple aryl rings reacted at the most electron-rich one (**38, 40, 41**, and **46**).

Scale-up of continuous flow electrosynthesis is usually achieved through scale-out (passing more material through the reactor) or numbering up using parallel reactors without the need to change reaction conditions[69]. Continuous production can be achieved if the electrochemical system is stable without electrode passivation or tubing blockage. To test the stability of the electrochemical system for the continuous production of aryl phosphonates, a solution of mesitylene and P(OEt)$_3$ in MeCN was mixed in-line with a solution of HBF$_4$ and H$_2$O in MeCN and passed through a reactor system consisting of two parallel electrochemical flow cells for 231 h (Fig. 3). The reaction produced 55.0 g (214 mmol) of the desired mesitylphosphonate **27** in 83% yield, which was even higher than the small-scale reaction (70%) and suggested that there was no reduction in reaction efficiency over time. The increased yield was probably due to the in situ mixing of HBF$_4$ and P(OEt)$_3$, the latter of which is known to undergo acid-promoted decomposition[70]. In fact, premixing of the reactants with HBF$_4$ caused yield reduction after a few hours. It is fortunate that the residence time in the reactor is short ($t_r$ = 75 s) and

in situ mixing is possible with continuous flow technology, showcasing the advantage of continuous flow electrosynthesis.

Diethyl mesitylphosphonate **27** could be hydrolyzed completely to phosphonic acid **47** (Fig. 3). One of the OEt groups of **27** could be replaced with different nucleophiles through a two-step process involving the intermediacy of ethyl mesitylphosphonochloridate to produce phosphinate **48**, phosphonamidate **49**, or mixed phosphonate **50**. Similar two-step procedures were also employed to convert **47** to triarylphosphine oxide **51** and P-heterocycles **52–54**.

**Mechanistic studies**. We next investigated the reaction mechanism. The cyclic voltammograms of P(OEt)$_3$, compound **1**, and HPO(OEt)$_2$ indicated that P(OEt)$_3$ ($E_{p/2}$ = 1.50 V vs SCE) was the easiest to oxidize among the three (Fig. 4a). The crude reaction mixture for the reaction of **1** with P(OEt)$_3$ with (standard conditions) or without H$_2$O (conditions I) was analyzed by $^{31}$P-NMR spectra (Fig. 4b). These experiments showed that P(OEt)$_3$ decomposed into several phosphorus species in the absence of H$_2$O. In contrast, under the standard conditions, the $^{31}$P-NMR spectrum was much cleaner with three major phosphorus species that were assigned to compound **2**, HPO(OEt)$_2$, and triethyl phosphate [OP(OEt)$_3$]. Since trialkyl phosphites underwent rapid acid-promoted hydrolysis to H-phosphonates[70], we thus conducted the phosphorylation reaction of **1** in the presence of both P(OEt)$_3$ and HPO(OEt)$_2$ without adding H$_2$O (conditions II and III). These reactions produced **2** in 58% yield in the presence of 2 equiv of HPO(OEt)$_2$ and a better yield of 78% with 0.2 equiv of HPO(OEt)$_2$ (Fig. 4b). Under these conditions, the $^{31}$P-NMR spectra of the crude reaction mixtures were similar to those under the standard conditions.

The electrolysis reaction of **1** in the presence of P(OEt)$_3$ and HPO(OnBu)$_2$ produced **2** in 51% yield with an only trace amount of **55** (Fig. 4c), suggesting that the PO(OR)$_2$ moiety of phosphonate product was derived from P(OR)$_3$ instead of HPO(OR)$_2$. Consistent with these results, $^{18}$O was not incorporated into **2** when the reaction was carried out with H$_2$$^{18}$O (Fig. 4d). These results pointed out that HPO(OR)$_2$ was critical for the C–H phosphorylation reaction. However, its exact role remained unclear. Finally, a kinetic isotope effect (KIE) of 1.0 was

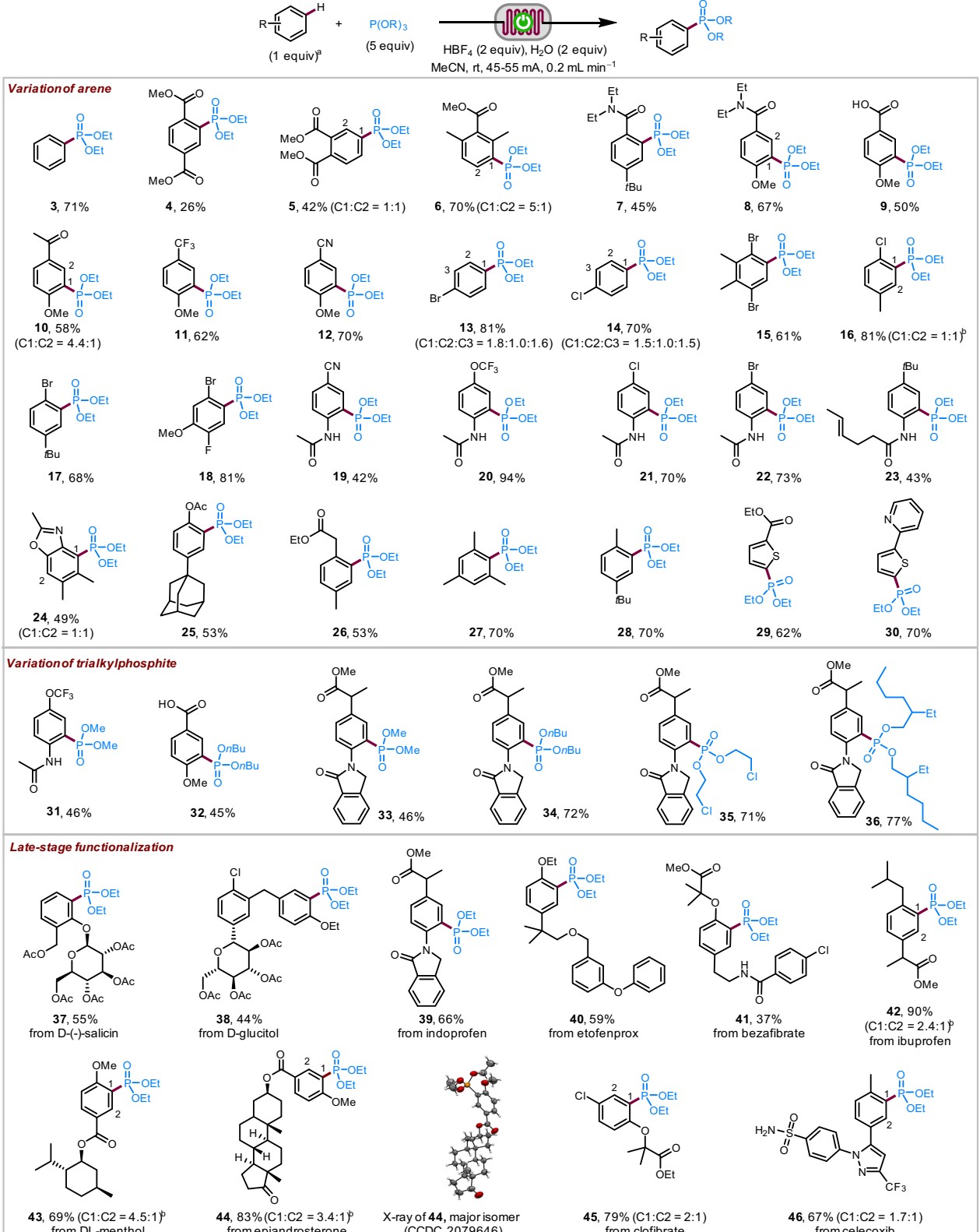

**Fig. 2 Reaction scope.** Reaction on 0.2 mmol of arene. All yields are isolated yields. [a]The ratio of regioisomers was determined by NMR analysis of crude reaction mixture. [b]Isomers were separable by chromatography. rt room temperature. Ac acetyl.

obtained from an intermolecular competition experiment with benzene and benzene-$d_6$ (Fig. 4e), suggesting that C(aryl)–H bond cleavage was not involved in the rate-limiting step of the electrochemical reaction.

Based on the above studies, a possible mechanism for the electrochemical C–H phosphorylation reaction is proposed (Fig. 4f). The reaction starts with the oxidation of trialkyl phosphite **56** on the anode to generate P-radical cation **57**, which

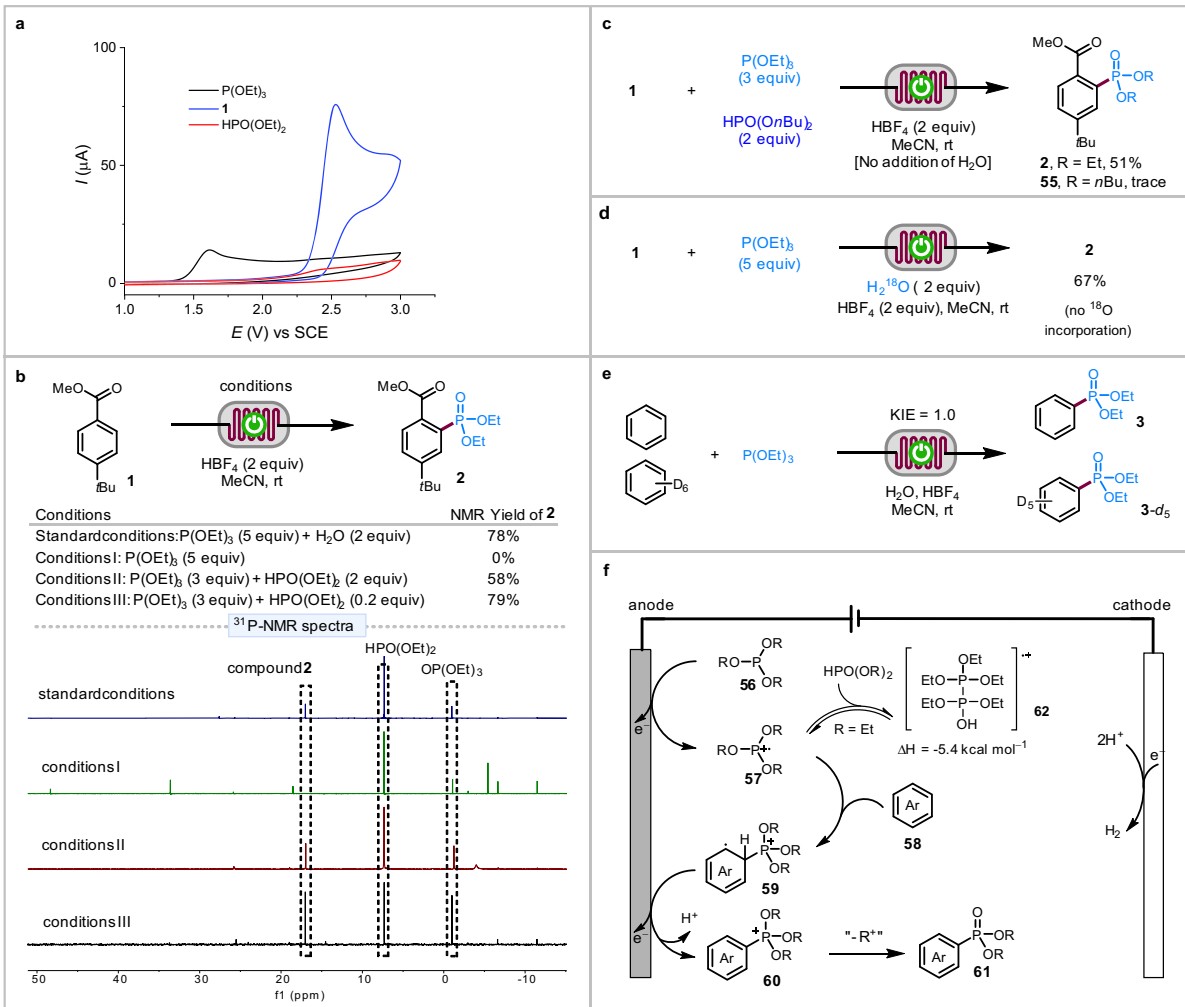

**Fig. 3 Reaction scale-up and product transformations.** Reaction conditions: [a]mesitylene (260 mmol), P(OEt)$_3$ (5 equiv), HBF$_4$ (2 equiv), H$_2$O (2 equiv), MeCN, rt, 231 h. [b]TMSBr, MeCN. [c]SOCl$_2$, DMF, reflux. [d]PhMgBr, THF. [e]HOCH$_2$CH$_2$CH$_2$NH$_2$, Et$_3$N, THF. [f](HOCH$_2$)$_2$CMe$_2$, Et$_3$N, THF. [g](MeNHCH$_2$)$_2$, Et$_3$N, THF. TMSBr trimethylsilyl bromide. THF tetrahydrofuran.

**Fig. 4 Mechanistic studies and proposal. a** Cyclic voltammograms. **b** Exploring the potential role of H$_2$O with $^{31}$P-NMR. The reaction mixture was treated with saturated aqueous NaHCO$_3$ before recording NMR spectra. **c** Electrolysis in the presence of P(OEt)$_3$ and HPO(OnBu)$_2$. **d** $^{18}$O-labeling experiment. **e** Kinetic isotope effect experiment. **f** Proposed mechanism. SCE saturated calomel electrode.

reacts with arene **58** to give distal radical cation **59**. The latter is oxidized further on the anode and then deprotonated to produce phosphonium **60**, which loses an alkyl group to a nucleophilic species in the reaction mixture such as H$_2$O or alcohol produced from the hydrolysis of P(OR)$_3$ or during the workup to afford the final phosphonate product **61**. Under these acidic conditions, protons, which are the most easily reduced species in the reaction mixture, accept electrons at the Pt cathode to generate H$_2$. In addition to promote hydrolysis of P(OR)$_3$, the acidic additive HBF$_4$ serves as the supporting electrolyte and a proton source for H$_2$ evolution, avoiding

unwanted cathodic reduction of electron-deficient species such as **60** and **61**. The $HPO(OR)_2$ formed in situ through the hydrolysis of $P(OR)_3$ likely forms reversibly with radical cation **57** an adduct **62**, which reduces the decomposition of **57** and buys more time for its reaction with the arene.

In summary, we have developed an electrochemical method for the direct C–H phosphorylation of arenes with trialkyl phosphites without the need for any molecular catalysts or conventional chemical oxidants. Key to the success is to include $H_2O$ as an additive and conduct the electrolysis with a continuous flow cell. The continuous flow electrosynthesis is compatible with electron-rich and electron-deficient arenes, providing rapid and scalable access to various aryl phosphonates from easily available materials.

## Methods

**Representative procedure for the electrochemical C–H phosphorylation.** The electrolysis was conducted using a flow electrolytic cell equipped with a graphite anode and a Pt cathode with an exposed surface area of 10 $cm^2$ and interelectrode distance of 0.25 mm. The solution containing arene substrate (0.05 M, 1 equiv), $P(OEt)_3$ (5 equiv), $HBF_4 \cdot Et_2O$ (2 equiv) and $H_2O$ (2 equiv) in dry MeCN was pushed using a syringe pump to pass through the flow electrolytic cell operated with a flow rate of 0.2 mL $min^{-1}$ and a constant current (45–55 mA). 4 mL of the outlet solution was collected once the reactor stabilized. The reaction was quenched with saturated $NaHCO_3$ and extracted with ethyl acetate. The organic extracts were combined and concentrated under reduced pressure. The residue was chromatographed through silica gel eluting with ethyl acetate/hexanes or methanol/dichloromethane to give the target product. All new compounds were fully characterized (see the Supplementary Methods).

## Data availability

The X-ray crystallographic coordinates for compound **44** have been deposited in the Cambridge Crystallographic Data Centre (CCDC), under deposition number 2079646 [www.ccdc.cam.ac.uk/data_request/cif]. The data supporting the findings of this study are available within the article and in the Supplementary Information file.

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

## Acknowledgements

We acknowledge NSFC (No. 21971213, H.-C.X.) and Fundamental Research Funds for the Central Universities for support.

## Author contributions

H.L., C.H., Y.-T.Z., Z.-Y.L., L.H.J., J.S. and S.Z. performed the experiments and analyzed the data. H.-C.X. designed and directed the project and wrote the manuscript.

## Competing interests

The authors declare no competing interests.
