## [Peer Review File · Nature Communications]

REVIEWER COMMENTS

Reviewer #1 (Remarks to the Author):

In this manuscript Hao Long et al report the electrochemical C-H phosphorylation in flow using phosphite precursors. With this protocol a broad variety of substrates were able to be transformed into the desired aryl phosphonate derivatives. Hao Long and co-workers were capable to address electron-poor substrates with high functional group tolerance such as esters, cyano, amides, halides, etc, and even late-stage functionalization was feasible. The approach on electron-poor substrates was a big challenge and has been demonstrated successfully. The electrochemical conversion in flow has been laid out properly and decent considerations have been made regarding the reaction mechanism and the reactivity of the phosphite precursor. Still some issues need to be stressed a bit.

Introduction:

- 1) First sentence: I would like to see more common organic chemical applications of aryl phosphonate compounds (2-3 sentences). You just have one small sentence. Which feels like you did not pay that much attention on the application. So, a bit more detail is highly desired.
- 2) You are saying in the introduction [...] arene or the phosphorus nucleophile [...] The term nucleophile is misleading. Because its resulting electron deficient intermediate will be most likely electrophilic. Just rephrase that
- 3) Later you are referring to reaction published where they used electron deficient arenes and they needed to use three electrode arrangement. In one of your Ref. they were using Co-cat in both compartments which means it is stable under oxidative and reductive conditions and no deposition can be seen. Please rethink that.

Results and discussion:

- 4) Can you explain this counterintuitive behaviour of the equivalents? 7eq or 3 eq does not work anymore. Have you ever tried to switch the excess components? Try to use the arene as excess since you proposed the phosphite is the one which is oxidized. An excess of trapping agent might be reasonable.
- 5) Can you explain the role of HBF₄ except for being a proton source for the hydrogen evolution at the platinum cathode? Did you use any additional supporting electrolyte such as Tetraalkylammonium salts? Tetrafluoroboric acid tends to be easily solvated which mean under these conditions it might also forms acid-base-salts which helps in terms of conductivity. Can you give any information on the cell potential of the reaction with TFA, AcOH, Sc(OTf)₃ and HBF₄?
- 6) Table 1: Have you ever tried TfOH. Which might be interfering with the phosphite and function as unproductive side reaction. But still interesting

Mechanistic studies:

- 7) Have you ever measured CV with the whole electrolyte system? HBF₄ and with or without water?
- 8) Conditions I: Have you seen the formation of the alkyl phosphonate instead? Which might occur under non-aqueous conditions with an acid present
- 9) [...] Since trialkyl phosphites underwent rapid acid-promoted hydrolysis to dialkyl phosphites, [...]. Isn't it rather a H-phosphonate than a dialkyl phosphite?
- 10) Regarding the hydrolysis of the phosphite. The oxygen will not be incorporated at all. Hence, this result was expected.
- 11) Have you quantify production of H-phosphonate with standard conditions? Might be interesting if its related to the amount of water? Because with conditions II you have been using the H-phosphonate in the same quantities like the amount of water with the standard conditions.
- 12) My suggestions:
 - a. Change the excess component 1 eq phosphite and 5 eq of arene
 - b. also try to use TfOH. Just as control reaction.
 - c. The role of the H-phosphonate is intriguing. Have you ever tried using catalytical amount of it? Would be very interesting to understand the role of this intermediate. Have you tried to monitor the process on-line by any means of NMR, IR.
- 13) Later you discuss the loss of the alkyl group by any nucleophilic species. In the Ref. 66 they mentioned that phosphonium salt is attacked either from water to produce MeOH or by the acid salt to produce the methyl ester. Might be an option here as well. Please discuss this in more detail

Reviewer #2 (Remarks to the Author):

In this manuscript, Xu and coworkers have developed a method for electrochemical C–H phosphorylation of arenes via the generation of a phosphorous radical cation. In this reaction, a trialkyl phosphite is electrochemically oxidized to generate the phosphorous radical cation which can react with the arene. Upon loss of an electron and a proton, dealkylation of the phosphonium occurs affording the final phosphonate product. This platform offers several advantages over previous C–H phosphorylation strategies: 1. Transition-metal catalyzed approaches require pre-installation of a directing group, limiting the arene scope of the transformation. 2. Undirected approaches usually proceed via oxidation of the arene substrate or the phosphorus nucleophile. However, the scopes of such processes are limited to electron-rich arenes due to the high oxidation potentials of the arene substrates or the reduced nucleophilicity of neutral phosphorus radicals. Xu and coworkers circumvented these challenges through the direct oxidation of trialkyl phosphites, reporting the seminal example of engaging phosphorus radical cations in C–H phosphorylation. The authors demonstrate that this method is synthetically useful through a broad scope, late-stage functionalization, product derivatization, and large scale synthesis.

Given the utility of these products and the clear advantages of this new protocol relative to state-of-the-art approaches, this manuscript will likely be of interest to the readership of Nature Communications and I recommend publication with minor revisions (*vide infra*).

In order to further improve this already strong manuscript, I recommend the authors consider the following suggestions that could improve this already strong manuscript:

- 1) The authors state that since there was no ¹⁸O incorporation in the product that HPO(OR)₂ is critical for the reaction and seem to indicate that water's role is forming this species. However, based on the NMR spectrum provided in Figure 4b, HPO(OR)₂ is formed in the absence of water. The authors should clarify this point and the role of water in the formation of HPO(OR)₂ as much as possible. These are interesting observations but currently almost completely unexplained in the manuscript. Have the authors examined whether trialkylphosphates could similarly promote the reaction?
- 2) The authors state that this reaction proceeds without the need for an oxidant. However, electrons must always be accepted in some way, in this case likely through cathodic proton reduction. The wording should be altered to state the transformation does not necessitate the use of a conventional chemical oxidant.
- 3) The authors screen several acid additives in the reaction. Is there a rationalization for why HBF₄ is the best acid for this transformation?
- 4) The clarity of Figure 4e would be improved by inclusion of the % ¹⁸O incorporation results.
- 5) The authors suggest that the increased yield observed in the scaleup reaction is probably due to the in situ mixing of HBF₄ and P(OEt)₃. Have the authors tested this in their standard conditions?
- 6) The manuscript states that the reaction performed is phosphorylation (C–O bond formation). However, since a C–P bond is formed, phosphorylation would be a more precise term.

Reviewer #3 (Remarks to the Author):

In this contribution, Xu and coworkers present a flow electrochemical procedure for the C-H phosphorylation of arenes. The authors claim that this is a novel transformation, applicable to electron-deficient arenes. Unfortunately, it appears that the authors have not carried out a literature background check. There are

many precedents on this type of electrochemical phosphorylation on electron poor aromatics. For example: 10.1080/10426507.2018.1540488, 10.1016/j.cattod.2016.06.001, 10.1007/BF00953100, 10.1080/10426507.2018.1541897, 10.1080/10426507.2016.1212051, 10.1080/10426507.2018.1540480. The chemistry presented is therefore not as novel as the authors suggest and, in opinion of this referee, it should not be accepted for publication in a top journal such as Nature Communications. It should be submitted to a more specialized journal.

There are some important additional issues listed below:

- The authors must cite all the literature mentioned above.
- Since the chemistry is known, the only novelty of this work is the adaptation of the electrolysis to continuous flow conditions. However, the development of the required conditions are not shared and are simply stated. It would be beneficial to the reader to understand the optimization process of these electrochemical parameters.

We thank the reviewers for taking their valuable time to check the manuscript and for giving constructive suggestions to improve it. We have worked tirelessly to address every comment of the reviewers. A point-to-point response to the comments is as following. The original reviewer comments are in blue.

Reviewer #1 (Remarks to the Author):

In this manuscript Hao Long et al report the electrochemical C-H phosphorylation in flow using phosphite precursors. With this protocol a broad variety of substrates were able to be transformed into the desired aryl phosphonate derivates. Hao Long and co-workers were capable to address electron-poor substrates with high functional group tolerance such as esters, cyano, amides, halides, etc, and even late-stage functionalization was feasible. The approach on electron-poor substrates was a big challenge and has been demonstrated successfully. The electrochemical conversion in flow has been laid out properly and decent considerations have been made regarding the reaction mechanism and the reactivity of the phosphite precursor. Still some issues need to be stressed a bit.

Response: We thank the reviewer for the positive comments.

Introduction:

1) First sentence: I would like to see more common organic chemical applications of aryl phosphonate compounds (2-3 sentences). You just have one small sentence. Which feels like you did not pay that much attention on the application. So, a bit more detail is highly desired.

Response: We have added more details and the first sentences now reads as the following. We would be happy to revise further if the review has additional suggestions on the wordings.

Aryl phosphorus compounds have wide applications in medicinal chemistry,¹ material science,² and catalysis as ligands and Lewis acid catalysts.^{3,4} In addition, brigatinib, which contains a phenylphosphine oxide motif and serves as an anaplastic lymphoma kinase (ALK) inhibitor, has been achieved commercial success for treating metastatic non-small-cell lung cancer (NSCLC).⁵

2) You are saying in the introduction [...] arene or the phosphorus nucleophile [...] The term nucleophile is misleading. Because its resulting electron deficient intermediate will be most likely electrophilic. Just rephrase that

Response: The sentence has been revised to the following:

In these reactions, either the arene or the phosphorus reagent is oxidized under...

3) Later you are referring to reaction published where they used electron deficient arenes and they needed to use three electrode arrangement. In one of your Ref. they were using Co-cat in both compartments which means it is stable under oxidative and reductive conditions and no deposition can be seen. Please rethink that.

Response: We have revised the sentence (see below). It is interesting that the dehydrogenative cross coupling can be achieved in both anodic and cathodic compartment. It is unclear how that works and why a divided cell is necessary if both electrodes can be used to promote product formation. More discussions on the refs of these authors are provided in responding the comments of reviewer 3.

Reactions of electron deficient arenes requires metal catalysts and impractical three electrode configuration in divided cells.^{42,43}

Results and discussion:

4) Can you explain this counterintuitive behaviour of the equivalents? 7eq or 3 eq does not work anymore. Have you ever tried to switch the excess components? Try to use the arene as excess since you proposed the phosphite is the one which is oxidized. An excess of trapping agent might be reasonable.

Response: For these reactions, an optimal concentration of $\text{P}(\text{OEt})_3$ is important for optimal results. With 3 equiv of $\text{P}(\text{OEt})_3$, a large portion of $\text{P}(\text{OEt})_3$ is hydrolyzed to $\text{HPO}(\text{OEt})_2$ because 2 equiv of H_2O is added to the system. In addition, some $\text{P}(\text{OEt})_3$ is protonated by the strong acid HBF_4 [pK_a in H_2O : $\text{HP}(\text{OMe})_3^+$, 2.6; HBF_4 , -0.3]. As a result, the amount of $\text{P}(\text{OEt})_3$ available is small leading to reaction failure. At higher concentration of $\text{P}(\text{OEt})_3$ such as 7 equiv, our hypothesis is that $\text{P}(\text{OEt})_3$ can now compete with the arene

to react with the radical cation $[(\text{EtO})_3\text{P}^{+\bullet}]$, leading to decomposition of $\text{P}(\text{OEt})_3$ and recovery of most arene starting material. With 7 equiv of $\text{P}(\text{OEt})_3$, we have observed the formation of $\text{EtPO}(\text{OEt})_2$, which likely arises from the reaction of $\text{P}(\text{OEt})_3$ with $[(\text{EtO})_3\text{P}^{+\bullet}]$. $\text{EtPO}(\text{OEt})_2$ is not observed under the standard conditions.

We agree that a higher concentration (excess) of arene is helpful for the arene to trap the P-radical cation $[(\text{EtO})_3\text{P}^{+\bullet}]$. But the arene is usually the more valuable component and it is better to use an excess of P-reagent for practical applications. We have conducted experiments with 5 equiv of arene and 1 equiv of $\text{P}(\text{OEt})_3$ with or without H_2O (2 equiv). The reaction failed in the presence of H_2O because $\text{P}(\text{OEt})_3$ is hydrolyzed to $\text{HPO}(\text{OEt})_2$. The reaction without H_2O provided **2** in 10% yield. These results under anhydrous conditions are consistent with early reports by A. N. Pudovik et al. (*Russ. Chem. Bull.* **1983**, 32, 566; ref 44). Pudovik and coworkers studied the electrochemical reactions of PhH, PhEt, and PhMe with $\text{P}(\text{OEt})_3$ under anhydrous conditions in a divided cell. They stated that “as large an excess as possible ArH” was needed to ensure good results (no details on the exact amount). Obviously, 5 equiv of compound **1** is not enough to obtain good yield for **2**. Hence, the use of excess of arene is not a practical solution.

5) Can you explain the role of HBF_4 except for being a proton source for the hydrogen evolution at the platinum cathode? Did you use any additional supporting electrolyte such as Tetraalkylammonium salts? Tetrafluoroboric acid tends to be easily solvated which mean under these conditions it might also formes acid-base-salts which helps in terms of conductivity. Can you give any information on the cell potential of the reaction with TFA, AcOH, $\text{Sc}(\text{OTf})_3$ and HBF_4 ?

Response: Role of HBF_4 . In addition to helping hydrogen evolution, HBF_4 facilitates the hydrolysis of $\text{P}(\text{OR})_3$ to produce $\text{HPO}(\text{OEt})_2$, which is critical for the success of the

electrochemical reaction. Without acid, the electron deficient cationic intermediate $[\text{ArP}(\text{OR})_3^+]$ or the product $\text{ArPO}(\text{OEt})_2$ can undergo reductive side reactions. The addition of 2 equiv of HBF_4 indeed reduces the cell potential from $> 50 \text{ V}$ (without HBF_4) to 2.9 V (standard conditions). So HBF_4 is also helpful to increase conductivity. For the reaction of compound **1**, replacing HBF_4 with $n\text{Bu}_4\text{NBF}_4$ (1 equiv) (cell potential = 5.4 V) leads to no formation of **2**. Hence, the role of HBF_4 is not just to increase conductivity. In the manuscript, we have stated the following.

In addition to promote hydrolysis of $\text{P}(\text{OEt})_3$, the acidic additive HBF_4 serves as the supporting electrolyte and is also helpful for H_2 evolution and avoiding unwanted cathodic reduction of electron-deficient species such as **60** and **61**.

The cell potentials of the reaction with 2 equiv of TFA, AcOH, TfOH, $\text{Sc}(\text{OTf})_3$, HBF_4 are 5.8 V , (increase over time), 2.5 V , 23 V , 2.9 V , respectively.

6) Table 1: Have you ever tried TfOH. Which might be interfering with the phosphite and function as unproductive side reaction. But still interesting

Response: Replacing HBF_4 with TfOH result in a lower yield of **2** (32%). TfOH is also corrosive to the graphite anode used. The results have been included in Table 1, entry 9.

Mechanistic studies:

7) Have you ever measured CV with the whole electrolyte system? HBF_4 and with or without water?

Response: We have added to the Supplementary Information the CV of the whole system consisting of compound **1** (1 equiv), $\text{P}(\text{OEt})_3$ (5 equiv), $\text{HBF}_4 \cdot \text{Et}_2\text{O}$ (2 equiv), and H_2O (2 equiv). A copy is displaced below (the blue trace in the figure below). The two oxidation peaks can be assigned to $\text{P}(\text{OEt})_3$ and compound **1** and $\text{HBF}_4 \cdot \text{Et}_2\text{O}$.

The CV of $\text{HBF}_4 \cdot \text{Et}_2\text{O}$ with or without water (1 equiv) has also been included in the Supplementary Information (also see below).

8) Conditions I: Have you seen the formation of the alkyl phosphonate instead? Which might occur under non-aqueous conditions with an acid present

Response: Yes, we have indeed observed $\text{EtPO}(\text{OEt})_2$ without adding H_2O . The peak at 33.7 ppm is assigned to $\text{EtPO}(\text{OEt})_2$ (See Fig. 4b of the manuscript). As we mentioned above, we have also observed the formation of $\text{EtPO}(\text{OEt})_2$ under the standard conditions but with 7 equiv of $\text{P}(\text{OEt})_3$.

9) [...] Since trialkyl phosphites underwent rapid acid-promoted hydrolysis to dialkyl phosphites, [...]. Isn't it rather a H-phosphonate than a dialkyl phosphite?

Response: Yes, $\text{HPO}(\text{OEt})_2$ is a H-phosphonate. But $\text{HPO}(\text{OEt})_2$ is called diethyl phosphite, likely because it behaves more like a trivalent phosphite than phosphonate. We have change dialkyl phosphites to H-phosphonates to avoid any confusion.

10) Regarding the hydrolysis of the phosphite. The oxygen will not be incorporated at all. Hence, this result was expected.

Response: We agree. The ^{18}O experiment also confirms that the $\text{PO}(\text{OR})_2$ moiety of the $\text{Ar-PO}(\text{OR})_2$ product did not originate from $\text{HPO}(\text{OR})_2$.

11) Have you quantify production of H-phosphonate with standard conditions? Might be interesting if its related to the amount of water? Because with conditions II you have been using the H-phosphonate in the same quantities like the amount of water with the standard conditions.

Response: We have used an internal standard (added after the electrolysis) to quantify the amount of H-phosphonate under the stand conditions. $\text{P}(\text{OPh})_3$ was added to the eluent of the cell and ^{31}P NMR was taken directly without workup. The results suggest that the yield for $\text{HPO}(\text{OEt})_2$ is about 62% from $\text{P}(\text{OEt})_3$. By following your suggestions, we have studied the reaction in the presence of different amount of $\text{HPO}(\text{OEt})_2$ (vide infra).

12) My suggestions:

a. Change the excess component 1 eq phosphite and 5 eq of arene

Response: We have done this experiment. Please see response to number 4) above.

b. also try to use TfOH. Just as control reaction.

Response: We have done this experiment. Please see response to number 6) above.

c. The role of the H-phosphonate is intriguing. Have you ever tried using catalytic amount of it? Would be very interesting to understand the role of this intermediate. Have you tried to monitor the process on-line by any means of NMR, IR.

Response: This is a great point on catalytic H-phosphonate. We have now studied the reactions on various amount of H-phosphonate. The reaction performed better in the presence of catalytic H-phosphonate. The best result was obtained with 0.2 equiv of $\text{HPO}(\text{OEt})_2$. The results with 0.2 equiv of $\text{HPO}(\text{OEt})_2$ have been included in Fig 4b of the manuscript.

Entry	X	Yield of 2 (%) ^a
1	0	30%
2	0.05	30%
3	0.1	55%
4	0.2	79%
5	0.5	69%
6	1.0	69%
7	2.0	58%

Consistent with these results, an optimal amount of H₂O is needed to adjust the concentration of P(OEt)₃ and HPO(OEt)₂ for best results (See below, the results have been included in Supplementary Table 1).

Screening on the amount of H₂O.^a

Entry	Deviation from standard conditions	Yield of 2 (%) ^b
1	none	70 ^c
2	H ₂ O (1 equiv)	26 (64)
3	H ₂ O (2.5 equiv)	65 (15)
4	H ₂ O (4 equiv)	26 (61)
5	H ₂ O (5 equiv)	0 (90)

^aStandard conditions: graphite anode, Pt cathode, interelectrode distance (0.25 mm), **1** (0.20 mmol), P(OEt)₃ (1.0 mmol), HBF₄•Et₂O (0.40 mmol), H₂O (0.40 mmol), MeCN (4.0 mL), flow rate = 0.20 mL min⁻¹, *t_r* (calculated residence time) = 75 s, constant current (55 mA), 3.4 F mol⁻¹.

^bDetermined by ¹H-NMR analysis using 1,3,5-trimethoxybenzene as the internal standard. Unreacted **1** was shown in brackets. ^cIsolated yield.

The results on the catalytic amount of HPO(OEt)₂ prompted us to apply these conditions to the difficult synthesis of **4** and to some of the substrates that are failed under the standard conditions (see below). But these conditions with catalytic HPO(OEt)₂ did not afford better results than our standard conditions for the difficult substrates. It is also unclear if these conditions with catalytic HPO(OEt)₂ are general and robust. But the results with compound **1** do show that it is possible to use less phosphorus reagents, e.g. 3 equiv of P(OEt)₃ and 0.2 equiv of HPO(OEt)₂. These conditions will be applied to our ongoing investigation on the synthetic applications of P-radical cations.

standard conditions: P(OEt)₃ (5 equiv), H₂O (2 equiv)

conditions A: P(OEt)₃ (3 equiv) + HPO(OEt)₂ (0.2 equiv)

standard conditions	26%	0%	0%	11%
conditions A	16%	0%	0%	15%

We did not monitor the process on-line. For a continuous flow electrochemical reaction, we assume that the reactions take place only in the cell. Once the solution flows out of the cell, no further product formation occurs. In a flow electrochemical cell, the conversion increase from entry to exit (See Figure below, reproduced from R. A. Green, R. C. D. Brown, D. Pletcher, *J. Flow Chem.* **2015**, 5, 31-36). For on-line analysis, we need to monitor the reaction at different positions of the reaction channel. We don't have the technology to do it.

13) Later you discuss the loss of the alkyl group by any nucleophilic species. In the Ref. 66 they mentioned that phosphonium salt is attacked either from water to produce MeOH or by the acid salt to produce the methyl ester. Might be an option here as well. Please discuss this in more detail

Response: We agree that nucleophilic species such as H₂O or alcohol should be the responsible for accepting the alkyl group. We have conducted additional experiments to address this point by detection of Bu₂O formation during the synthesis of **34**. The use of HBF₄•Et₂O as an additive prevented us to study with P(OEt)₃. Hence, we studied the reaction of P(OBu)₃. The *n*Bu group may be picked up by H₂O or *n*BuOH (generated from P(OBu)₃ and H₂O) to produce *n*BuOH and *n*Bu₂O respectively. We have observed formation of *n*Bu₂O by GC analysis of the reaction mixture, suggesting *n*BuOH is at least one of the species to accept the alkyl group. This experiment has been included in the Supplementary Information. We added the following discussion to the manuscript: ...which loses an alkyl group to a nucleophilic species in the reaction mixture such as H₂O or alcohol produced from hydrolysis of P(OR)₃ or during the workup to afford the final phosphonate product **61**.

Reviewer #2 (Remarks to the Author):

In this manuscript, Xu and coworkers have developed a method for electrochemical C–H phosphorylation of arenes via the generation of a phosphorous radical cation. In this reaction, a trialkyl phosphite is electrochemically oxidized to generate the phosphorous radical cation which can react with the arene. Upon loss of an electron and a proton, dealkylation of the phosphonium occurs affording the final phosphonate product. This platform offers several advantages over previous C–H phosphorylation strategies: 1. Transition-metal catalyzed approaches require pre-installation of a directing group, limiting the arene scope of the transformation. 2. Undirected approaches usually proceed via oxidation of the arene substrate or the phosphorus nucleophile. However, the scopes of such processes are limited to electron-rich arenes due to the high oxidation potentials of the arene substrates or the reduced nucleophilicity of neutral phosphorus radicals. Xu and coworkers circumvented these challenges through the direct oxidation of trialkyl phosphites, reporting the seminal example of engaging phosphorus radical cations in C–H phosphorylation. The authors demonstrate that this method is synthetically useful through a broad scope, late-stage functionalization, product derivatization, and large scale synthesis.

Given the utility of these products and the clear advantages of this new protocol relative to state-of-the-art approaches, this manuscript will likely be of interest to the readership of Nature Communications and I recommend publication with minor revisions (*vide infra*).

Response: We thank the reviewer for the positive recommendation.

In order to further improve this already strong manuscript, I recommend the authors consider the following suggestions that could improve this already strong manuscript:

1) The authors state that since there was no ^{18}O incorporation in the product that $\text{HPO}(\text{OR})_2$ is critical for the reaction and seem to indicate that water's role is forming this species. However, based on the NMR spectrum provided in Figure 4b, $\text{HPO}(\text{OR})_2$ is formed in the absence of water. The authors should clarify this point and the role of water in the formation of $\text{HPO}(\text{OR})_2$ as much as possible. These are interesting observations but currently almost completely unexplained in the manuscript. Have the authors examined whether trialkylphosphates could similarly promote the reaction?

Response: We thank the reviewer for these excellent comments. Aqueous NaHCO_3 was added to quench the reaction before taking NMR. Hence, $\text{HPO}(\text{OEt})_2$ was observed in all conditions. We have now clarified this in the Fig caption. Detailed procedures for the mechanistic studies have been provided in the Supplementary Information.

H_2O is added to react with $\text{P}(\text{OEt})_3$ to generate $\text{HPO}(\text{OEt})_2$. $\text{HPO}(\text{OEt})_2$ is critical for a good yield. This is clear from the results of Figure 4d. The question is why $\text{HPO}(\text{OEt})_2$ is helpful. We are very interested in this observation as it provides us an effective method to use $[(\text{EtO})_3\text{P}^{+\cdot}]$ in organic synthesis. It is interesting that both a catalytic amount of $\text{HPO}(\text{OEt})_2$ and $\text{OP}(\text{OEt})_3$ are helpful (See below). We have determined that under the electrochemical conditions $\text{OP}(\text{OEt})_3$ is stable and does not convert to $\text{HPO}(\text{OEt})_2$. Our current hypothesis is that $\text{HPO}(\text{OEt})_2$ and $\text{OP}(\text{OEt})_3$ can interact reversibly with the radical cation of $[(\text{EtO})_3\text{P}^{+\cdot}]$, which increases the lifetime of the radical cation and buys more time for its reaction with the arene. In other words, the adduct between $\text{HPO}(\text{OEt})_2$ and $[(\text{EtO})_3\text{P}^{+\cdot}]$ serves as a reservoir for the radical cation $[(\text{EtO})_3\text{P}^{+\cdot}]$. Computational calculations suggest both the reactions of $\text{HPO}(\text{OEt})_2$ and $\text{OP}(\text{OEt})_3$ with $[(\text{EtO})_3\text{P}^{+\cdot}]$ is exothermic but reversible, providing supporting to our hypothesis. Consistent with this hypothesis, an optimal concentration of $\text{HPO}(\text{OEt})_2$ or $\text{OP}(\text{OEt})_3$ is needed for the best results.

The reaction efficiency decreases at high concentration of $\text{HPO}(\text{OEt})_2$ or $\text{OP}(\text{OEt})_3$ because the equilibrium concentration of the radical cation $[(\text{EtO})_3\text{P}^{+\bullet}]$ becomes low, which reduces its chance to react effectively with arenes. We have added adduct formation to Fig 4f and the following discussion on the role of the $\text{HPO}(\text{OR})_2$ to the text: **The $\text{HPO}(\text{OR})_2$ formed in situ through the hydrolysis of $\text{P}(\text{OR})_3$ likely forms reversibly with radical cation **59** an adduct **62**, which reduces the decomposition of **59** and buys more time for its reaction with the arene.**

Entry	X	Yield of 2 (%) ^a
1	0	30%
2	0.2	66%
3	1	56%
4	2	58%

Entry	X	Yield of 2 (%) ^a
1	0	30%
2	0.05	30%
3	0.1	55%
4	0.2	79%
5	0.5	69%
6	1.0	69%
7	2.0	58%

2) The authors state that this reaction proceeds without the need for an oxidant. However, electrons must always be accepted in some way, in this case likely through cathodic proton reduction. The wording should be altered to state the transformation does not necessitate the use of a conventional chemical oxidant.

Response: We have changed “chemical oxidants” to “conventional chemical oxidants” and “oxidant-free” to “external oxidant-free”.

3) The authors screen several acid additives in the reaction. Is there a rationalization for why HBF₄ is the best acid for this transformation?

Response: HBF₄, which contains a non-nucleophilic anion and is a strong acid, serves as electrolyte and promotes H₂ evolution and the hydrolysis of some of P(OEt)₃ to HPO(OEt)₂. Acids such as TFA and AcOH with nucleophilic anions can compete with the arene to react with the radical cation derived from P(OEt)₃. As a result, no product **2** was formed with TFA and AcOH. TfOH is known to cause the decomposition of P(OEt)₃ (ref 68) and is also corrosive to the graphite anode. The reaction with TfOH afforded 32% of **2** with 50% of **1** unreacted. The cell potentials of the reaction with 2 equiv of TFA, AcOH, TfOH, Sc(OTf)₃, HBF₄ are 5.8 V, 16–35 V (increase over time), 2.5 V, 23 V, 2.9 V, respectively. The cell potentials for the reactions with weak acid AcOH and Lewis acid Sc(OTf)₃ are high, suggesting low conductivity. With AcOH, TFA, or Sc(OTf)₃ as the additive, more than 80% of arene **1** is left unreacted, suggesting that decomposition of P(OEt)₃ is the major reaction in the cell. The cell potentials have been included in the Supplementary Information.

4) The clarity of Figure 4e would be improved by inclusion of the % ^{18}O incorporation results.

Response: We have included the results (no ^{18}O incorporation) in Figure 4d.

5) The authors suggest that the increased yield observed in the scaleup reaction is probably due to the in situ mixing of HBF_4 and $\text{P}(\text{OEt})_3$. Have the authors tested this in their standard conditions?

Response: For the continuous flow synthesis, the scaleup is achieved by using parallel reactors and by passing more materials through the reactor. The conditions for the scaleup are the same with the standard conditions except the in-situ mixing. The reaction under standard conditions (small scale) is run for 20 min. You can view the scale up as the continuation of standard conditions (many 20 min). The yield of the large scale is better than the small scale suggesting that yield for every 20 min should be better than the small scale. We did not test this in situ mixing for other substrates.

6) The manuscript states that the reaction performed is phosphorylation (C–O bond formation). However, since a C–P bond is formed, phosphonylation would be a more precise term.

Response: We have changed phosphorylation to phosphonylation. We agree that phosphonylation clearly describes the reaction since the products are phosphonates. Phosphorylation refers to the attachment of a phosphoryl group (PO_3^{2-}) to a molecular (e.g. X) through P-X linkage, not P-O-X linkage. For example, protein phosphorylation occurs on OH and NH_2 groups of the proteins to form O- PO_3^{2-} or NH- PO_3^{2-} moieties. Both phosphorylation and phosphonylation are used in the literature to describe C-P bond formation. The transformations of Ar-H to Ar- $\text{PO}(\text{OR})_2$ are referred to as phosphorylations in the literature (e.g. Yu, J.-Q. et al. Pd(II)-Catalyzed Phosphorylation of Aryl C–H Bonds. *J. Am. Chem. Soc.* **2013**, *135*, 9322; Ackermann, L. et al. Electrocatalytic C-H phosphorylation through nickel(III/IV/II) catalysis. *Chem* **2021**, *7*, 1379).

Reviewer #3 (Remarks to the Author):

In this contribution, Xu and coworkers present a flow electrochemical procedure for the C-H phosphorylation of arenes. The authors claim that this is a novel transformation, applicable to electron-deficient arenes. Unfortunately, it appears that the authors have not carried out a literature background check. There are many precedents on this type of electrochemical phosphorylation on electron poor aromatics. For example: 10.1080/10426507.2018.1540488, 10.1016/j.cattod.2016.06.001, 10.1007/BF00953100, 10.1080/10426507.2018.1541897, 10.1080/10426507.2016.1212051, 10.1080/10426507.2018.1540480.

The chemistry presented is therefore not as novel as the authors suggest and, in opinion of this referee, it should not be accepted for publication in a top journal such as Nature Communications. It should be submitted to a more specialized journal.

Response: We thank the reviewers for providing these references, which are summarized in Table R1 as refs I to VI (see below). *We believe without doubt that these previous works do not reduce the novelty of our work. On the contrary, they highlight the advantage of our method and increase the novelty of our work.*

Two of these references (refs I and II) have already been cited in our original submission. We have now added all the refs as refs 42-47 in the manuscript. These references are from recent work of Y. Budnikova and coworkers, except for ref. III. The refs V and VI are review papers on their own work and will not be discussed further. Ref. IV only deals with the phosphorylation of caffeine. The reactions in ref IV are carried out in a divided cell using three-electrode system with Pd(OAc)₂ or AgOAc or Bipy₃Ni(BF₄)₂ (cat.) as the catalyst. No information is given on the type of electrodes employed or the amount of substrate and catalyst used. Ref. III is one of the pioneering studies of electrochemical C-H phosphorylation and dealt with the reactions of benzene, ethylbenzene and toluene. These reactions employed P(OR)₃ as the phosphorylation reagent and required a divided cell and large excess of arenes (The authors stated “as large excess as possible of ArH over the trialkylphosphite” without providing the exact amount).

Refs I and II deal with the same substrates but with different metal catalysts, CoCl₂bpy or a bimetallic system MnCl₂bpy/Ni(BF₄)₂bpy. These reactions are shown to be compatible with electron-deficient arenes such as PhCN and PhNO₂ but required metal catalyst(s), divided cells, three-electrode system under constant potential conditions, and a large amount of salt (saturated solution of Et₄NBF₄ in MeCN). In these divided cells equipped with a porous ceramic separator or ionic exchange membranes, the mixing of the organic

solution in the anodic and cathodic compartments is difficult to avoid overtime. The use of a separator brings high resistance, necessitating the use of high concentration of salts. For the reactions under constant potential, the current reduces overtime, and it usually takes a very long time for the reaction to finish. As a result, these conditions make the reaction scale up very difficult. For the reactions in refs II to afford good yields, the reaction components need to be mixed for one day before electrolysis. Electrolysis immediately after mixing the reactants afforded yields all below 40%, making these reactions even more challenge to carry out. These references highlight the challenges of electrochemical arene C-H phosphorylation.

The reactions in simple undivided cells without metal catalysts are highly desirable but very challenging. Lei and coworkers have reported electrochemical C-H phosphorylation reactions using undivided cells through direct electrolysis (ref 38, *Chem. Commun.* **2019**, 55, 4230) or indirect electrolysis with Mn(OAc)₂ as the catalyst (ref 39, *ACS Catal.* **2021**, 4295). These reactions dealt with electron-rich heteroarenes such as imidazo[1,2-a]pyridines, (benzo)thiophenes, and (benzo)furans. The reaction efficiency of *p*-xylene (ref 38), an arene that is not electron-deficient, reduces significantly (38% yield). These results further underline the challenges of developing efficient, practical, and metal catalyst-free arene C-H phosphorylation reactions in undivided cells. Advantageously, our reactions show broad scope, requires no metal catalysts, are carried out in undivided flow cells under constant current conditions and scalable.

Some additional comments on ref. I are as following.

1. All the 28 references cited in the paper are self-citations. It is rather unusual for a paper published in 2019 to cite only their own work.
2. There is a lack of details on the reactions.
 - a) No information on the scale of the reactions is given.
 - b) The reactions are conducted using a three-electrode system, but no information is given on the potentials applied.
 - c) The yields are referred to those of the main product as suggested by Table 1, but no information is given on the regioisomeric ratios for the reactions of PhNMe₂, PhCN and PhNO₂. In their other paper (ref. II) with Mn/Ni based catalysts, the regioisomeric ratios were between 1:1 to 1:4.
 - d) This work shows that C-H phosphorylation of arenes is successful in both the anodic and cathodic compartment. The authors suggested in Scheme 1 that the cathodic process involved the sequential loss of two hydrogen atoms to give the final product

but give no explanation in the text. But the loss of a hydrogen atom from $[\text{Co}^{2+}\text{LS-H}]^{\ominus}$ cannot give a P-radical and $[\text{Co}^{2+}\text{L}]$ as shown in Scheme 1.

Table R1. Summary of the references provided by reviewer 3	
Ref I	Y. H. Budnikova et al., Phosphorus, Sulfur, and Silicon and the Related Elements 2019, 194, 506, DOI: 10.1080/10426507.2018.1540488 Reaction scope:  R = H, NMe₂, CN, NO₂ R = 6-Me, 7-Me P-reagent: HPO(OEt)₂ Conditions: Divided cell, three electrode system (constant potential), CoCl₂bpy (cat.), saturated Et₄NBF₄ in MeCN. No information on reaction scale and potentials applied.
Ref II	Y. H. Budnikova et al., Catalysis Today 2017, 279, 133, DOI: 10.1016/j.cattod.2016.06.001. Reaction scope:  R = H, NMe₂, CN, NO₂ R = 6-Me, 7-Me P-reagent: HPO(OR)₂ (R = Et, iPr, nBu) Conditions: Divided cell, three electrode system (constant potential), MnCl₂bpy/Ni(BF₄)₂bpy (cat.), Et₄NBF₄ in MeCN (anodic compartment), saturated Et₄NBF₄ and PyHBF₄ in MeCN (cathodic compartment). Electrolysis one day after the mixing of the substrates provided better yields.
Ref III	A. N. Pudovik et al., Russ. Chem. Bull. 1983, 32, 566, DOI:10.1007/BF00953100 Reaction scope: PhH, PhEt, PhMe P-reagent: P(OR)₃ (R = Et, Pr, Bu) Conditions: divided cell, 4-5 mA/cm², 0.1 M NaClO₄, large excess of arene. No information on reaction scale.
Ref IV	Y. H. Budnikova et al., Phosphorus, Sulfur, and Silicon and the Related Elements 2019, 194, 343, DOI:10.1080/10426507.2018.1541897 Reaction scope: Caffeine P-reagent: HPO(OR)₂ (R = Et, iPr, nBu) Conditions: divided cell, three electrode system (constant potential), Pd(OAc)₂ or AgOAc or Bipy₃Ni(BF₄)₂ (cat.). No information on reaction scale.
Ref V	Y. H. Budnikova et al., Phosphorus, Sulfur, and Silicon and the Related Elements 2016, 191, DOI:149110.1080/10426507.2016.1212051 Microreview on the authors' own work

Ref VI	Y. H. Budnikova et al., Phosphorus, Sulfur, and Silicon and the Related Elements 2019 , 194 , 415, DOI:10.1080/10426507.2018.1540480 Review on the authors' own work.
--------	--

There are some important additional issues listed below:

- The authors must cite all the literature mentioned above.

Response: The references have been cited as 42-47.

- Since the chemistry is known, the only novelty of this work is the adaptation of the electrolysis to continuous flow conditions. However, the development of the required conditions are not shared and are simply stated. It would be beneficial to the reader to understand the optimization process of these electrochemical parameters.

Response: Our work is not just simple adaption of the electrolysis in continuous flow as discussed above and represents significant advances in arene C-H phosphorylation. Divided cells, constant potential conditions, and metal catalysts are used because simple conditions involving undivided cells, constant current and metal-free do not work, not because people prefer the complicated setup and inconvenient conditions. We make it work through innovation on technology, mechanism, and reaction conditions. Simply moving the reactions in the references listed by the reviewer to flow will not work. The reactions of Budnikova and coworkers proceed through addition of P-radicals to the arenes. To achieve efficient reactions in undivided cells without metal catalysts, we resort to a different mechanism involving reactions of P-radical cations with arenes. Ref III follows a similar mechanism but required a divided cell and "as large an access as possible ArH" as the authors stated. Also, the reactions in ref III are conducted under dry conditions because water reacts with P-radicals and is considered to have negative effect on the yield (see Eq 3, page 569 of ref III). As we have demonstrated, simply moving these dry conditions to flow do not work. In contrast, we have added water to facilitate the reactions and are able to achieve efficient synthesis with 1 equiv of arenes, which is the valuable coupling partner.

More information on the reaction optimization has been added to the Supplementary Table 1.

REVIEWERS' COMMENTS

Reviewer #1 (Remarks to the Author):

In this manuscript Hao Long et al report the electrochemical C-H phosphorylation in flow using phosphite precursors. My suggestions and comments have been implemented successfully. Some minor comments are still open for discussion (vida infra). Overall, this new electrochemical protocol for the C-H phosphorylation in flow will be most likely of interest for the academic community of Nature Communications and their readership. This approach can be considered novel as a two-electrode setup is used under galvanostatic conditions without the need for transition metal catalysts and even electron-deficient substrates are accessible. Additionally, the use of flow technology demonstrates the scalability up to 55 g scale (impressive!) and emphasize the sustainability aspect of this electrosynthesis as well. Hence, I support the acceptance of this publication with minor revisions.

Question 5:

"Without acid, the electron deficient cationic intermediate $[\text{ArP}(\text{OR})_3^+]$ or the product $\text{ArP}(\text{OEt})_2$ can undergo reductive side reactions"

– Can you explain this? Why does the use of acid prevent reductive side reactions? Maybe it is related to the double layer polarity which is different if you compare the n-butylammonium salt with HBF_4 close to the positive polarized cathode.

Mechanistic Studies:

"The $\text{HPO}(\text{OR})_2$ formed in situ through the hydrolysis of $\text{P}(\text{OR})_3$ likely forms reversibly with radical cation 59 an adduct 62, which reduces the decomposition of 59 and buys more time for its reaction with the arene."

– Isn't it rather 57 than 59?

– It is reasonable to state that this intermediate 62 will act as reservoir.

Reviewer #2 (Remarks to the Author):

I am completely satisfied by the revisions to this manuscript. I think the authors did an excellent job responding to the comments of all 3 reviewers. I found the arguments, text clarifications, and new data compelling and recommend publication without further change.

A point-to-point response to the comments of reviewers is as following. The original reviewer comments are in black, and our responses are in blue.

Reviewer #1 (Remarks to the Author):

In this manuscript Hao Long et al report the electrochemical C-H phosphorylation in flow using phosphite precursors. My suggestions and comments have been implemented successfully. Some minor comments are still open for discussion (*vida infra*). Overall, this new electrochemical protocol for the C-H phosphorylation in flow will be most likely of interest for the academic community of Nature Communications and their readership. This approach can be considered novel as a two-electrode setup is used under galvanostatic conditions without the need for transition metal catalysts and even electron-deficient substrates are accessible. Additionally, the use of flow technology demonstrates the scalability up to 55 g scale (impressive!) and emphasize the sustainability aspect of this electrosynthesis as well. Hence, I support the acceptance of this publication with minor revisions.

Response: We thank the reviewer for taking time to review the manuscript again and for the positive recommendation.

Question 5:

“Without acid, the electron deficient cationic intermediate $[\text{ArP}(\text{OR})_3^+]$ or the product $\text{ArP}(\text{OEt})_2$ can undergo reductive side reactions”

- Can you explain this? Why does the use of acid prevent reductive side reactions? Maybe it is related to the double layer polarity which is different if you compare the n-butylammonium salt with HBF_4 close to the positive polarized cathode.

Response: We thank the reviewer for this question. At the Pt cathode, the most easily reduced species accepts electrons from the electrode. With an acid present, the most easily reduced species is proton. Hence protons undergo reduction to H_2 when HBF_4 is included as an additive. However, without an added acid, the proton concentration in the reaction mixture is low, and other electron-deficient species such as $[\text{ArP}(\text{OR})_3]^+$ or the product $\text{ArP}(\text{OEt})_2$ become the most easily reduced species and may undergo to reduction

at the cathode leading to their decomposition. We have modified the discussion on the role of acid in the text to the following.

Under these acidic conditions, protons, which are the mostly easily reduced species in the reaction mixture, accept electrons at the Pt cathode to generate H₂. In addition to promote hydrolysis of P(OR)₃, the acidic additive HBF₄ serves as the supporting electrolyte and a proton source for H₂ evolution, avoiding unwanted cathodic reduction of electron-deficient species such as **60** and **61**.

Mechanistic Studies:

“The HPO(OR)₂ formed in situ through the hydrolysis of P(OR)₃ likely forms reversibly with radical cation **59** an adduct **62**, which reduces the decomposition of **59** and buys more time for its reaction with the arene.”

- Isn't it rather **57** than **59**?

Response: We thank the reviewer for pointing out the typo. It is indeed **57**. We have corrected it to show as **57** (see below).

The HPO(OR)₂ formed in situ through the hydrolysis of P(OR)₃ likely forms reversibly with radical cation **57** an adduct **62**, which reduces the decomposition of **57** and buys more time for its reaction with the arene.

- It is reasonable to state that this intermediate **62** will act as reservoir.

Response: We thank the reviewer for agreeing on our proposal.

Reviewer #2 (Remarks to the Author):

I am completely satisfied by the revisions to this manuscript. I think the authors did an excellent job responding to the comments of all 3 reviewers. I found the arguments, text clarifications, and new data compelling and recommend publication without further change.

Response: We thank the reviewer for taking time to review the manuscript again and for the positive recommendation.